# FAIR OUT-OF-DISTRIBUTION DETECTION

## ABSTRACT

Out-of-Distribution (OOD) detection prevents models from misclassifying OOD data that fall outside the in-distribution (ID) classes as ID categories. However, existing OOD detection methods ignore a critical metric, i.e., fairness metric. This oversight could result in unreliable predictions due to sensitive attributes in the data. To fill this gap, we introduce a novel and challenging problem termed *Fair OOD Detection* in this paper, which simultaneously considers OOD detection and bias induced by *Fairness Confusion* (FC) caused by sensitive attributes and their induced *Feature Shifts* (FS). Furthermore, we propose a novel metric termed Fair-OOD to identify FC phenomena in OOD detection, and a theoretically guaranteed semi-supervised solution named *Predictive Adaptive Calibration* (PACT) to simultaneously enhance OOD detection capability, ensure fairness, and mitigate FC without requiring the label of sensitive attribute for OOD data. Extensive experiments demonstrate that: (a) Fair-OOD can identify FC issues in models that existing fairness metrics fail to detect; (b) PACT effectively improves OOD detection performance while eliminating both FC and unfairness issues.

## 1 INTRODUCTION

Machine learning (ML) classification models trained under the closed-world assumption inevitably encounter out-of-distribution (OOD) inputs from unknown categories in real-world applications. As a result, OOD detection serves as a critical safety mechanism by identifying OOD samples while maintaining accurate classification on in-distribution (ID) data, thereby ensuring reliable model performance beyond the training distribution (Wang et al., 2025). Existing research in OOD detection primarily focuses on two directions: developing more efficient fine-tuning strategies (Hendrycks et al., 2019; Liu et al., 2020; Wang et al., 2023a), and designing effective scoring functions that better distinguish ID from OOD data (Hendrycks & Gimpel, 2017; Wang et al., 2023b).

Despite the notable progress in building robust and trustworthy ML systems, we observe that concurrent OOD detection approaches overlook another relative but important constraint, i.e., the fairness metric (Jung et al., 2024; Hardt et al., 2016), which requires equal predictions across different groups decided by the *sensitive attribute*. As illustrated in Fig. 1, the major and minor sample groups governed by the sensitive attribute (e.g., image background) exhibit similar distribution division patterns to those in OOD detection tasks, i.e., ID and OOD distributions (Wang et al., 2025). Unfortunately, as recent researchers have been aware of the frequent co-occurrence of the sensitive attribute within realistic data (Nam et al., 2020), this *relative but divergent* variable tends to confuse distinguishing between ID and OOD data, thereby undermining the performance of OOD detection. To be specific, we characterize this unexpected but commonly existed phenomenon as *Fairness Confusion* (FC), featuring the negative impact of the sensitive attribute from two folds. First, as illustrated in Fig. 1, the sensitive attributes themselves might yield direct, biased impact on OOD detection (e.g., the OOD horse is misclassified as a tiger or sika deer due to background). Moreover, sensitive attributes may further induce *Feature Shifts* (FS) and yield indirect impact by yielding feature disparities similar to the disparities between ID and OOD distributions. Leaving this issue unresolved undermines the reliability of OOD detection, making its resolution essential.

To migrate the fairness confusion, we thereby introduce a novel and challenging problem termed *Fair OOD Detection* (cf. Problem 1) in this paper, which simultaneously considers both the sensitive attributes of data and the *Feature Shifts* (FS) induced by these attributes. To the best of our knowledge, our work presents the first study on constructing fairness-aware OOD detection framework. Furthermore, as our analysis reveals that the fairness metrics fails to capture the impact of FS, their

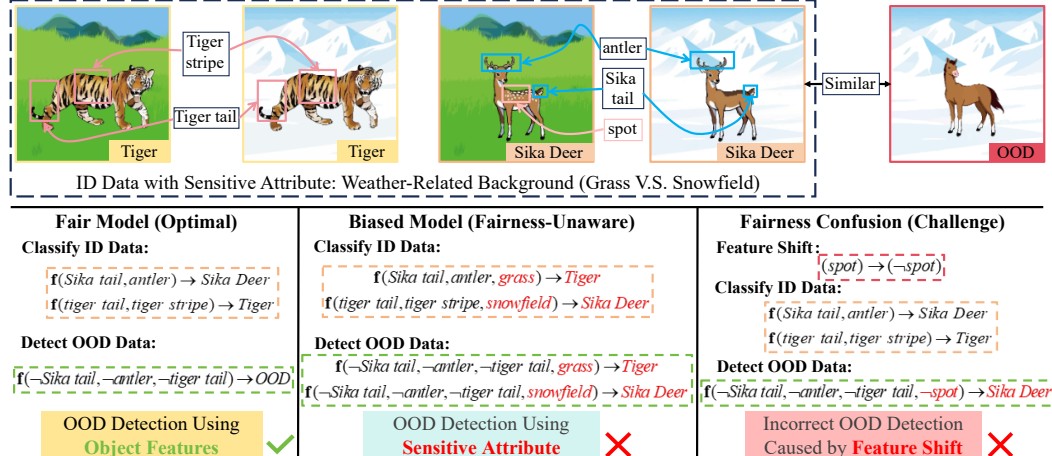

Figure 1: Comparison of Fair Models (optimal), Biased Models (fairness-unaware), and Fairness Confusion (challenge) in OOD Detection. The symbol ¬ denotes logical NOT and **f** represents the model. Red font indicates erroneous evidence that should not be relied upon, or incorrect results. In OOD detection, besides the unfairness caused by sensitive attributes, a distinct fairness confusion problem emerges from feature shifts induced by sensitive attributes. However, current OOD detection literature overlooks both fairness and fairness confusion, a gap that motivates our work.

direct adaptation is limited. As shown in our example in Fig. 1, even when a model satisfies existing fairness metrics with respect to sensitive attributes themselves (e.g., backgrounds), it may still produce erroneous detection outcomes due to FS. To handle this challenge, we subsequently introduce the Fair-OOD metric (cf. Definition 1) to identify the FC phenomenon in OOD detection. It works by evaluating the prediction consistency of a model under varying sensitive attributes and FS conditions.

To address our proposed fair OOD detection problem, we subsequently propose a novel semi-supervised approach termed *Predictive Adaptive Calibration* (PACT), which is designed to achieve three key objectives: (a) enhancing the OOD detection capability of model, (b) ensuring fairness, and (c) mitigating FC. Specifically, PACT employs *Feature Distribution Regularization* (FDR) to align feature distributions within the same class while increasing the divergence of distributions across different classes. Concurrently, it utilizes *Predictive Distribution Calibration* (PDC) to amplify the discrepancy between the prediction distributions of ID and OOD data when the label of sensitive attribute in OOD data are unavailable. Furthermore, we conduct theoretical analyses and provide formal guarantees for both FDR and PDC. We summarize our main contributions as follows:

- We first incorporate the consideration of fairness issues in OOD detection and introduce a novel problem, Fair OOD Detection, by considering the issue, i.e., *Fairness Confusion* (FC).

- We introduce a novel metric, Fair-OOD, to identify the FC issue in OOD detection.

- We propose a novel algorithm, *Predictive Adaptive Calibration* (PACT) (cf. Algorithm 1), which simultaneously ensures model fairness, enhances OOD detection capabilities, and mitigates FC issue. And we provide theoretical guarantees for it.

- Extensive experiments on real-world datasets demonstrate that: (a) Our proposed metric, Fair-OOD, effectively identifies FC problems; and (b) Our proposed method, PACT, improves OOD detection performance while eliminating both FC and unfairness issues.

## 2 EMPIRICAL STUDY: FAIRNESS CONFUSION

In this section, we conduct a case study to investigate the following two questions:

- How does sensitive attribute affect OOD detection?

- How does feature shift (FS) affect OOD detection?

## 2.1 NOTATIONS

Let $\mathcal{X}$ denote the feature space and $\mathcal{Y} = \{1, \cdots, c\}$ denote the ID label space. To facilitate formal analysis, we introduce a binary variable $Z \in \mathcal{Z} = \{i, o\}$ to indicate ID ($Z = i$) or OOD ($Z = o$) for $X \in \mathcal{X}$. Furthermore, as discussed in the introduction, the model may be influenced by sensitive attributes and their induced feature shifts. To formalize this, we denote the sensitive attribute and feature shift by the variables $A$ and $S$,

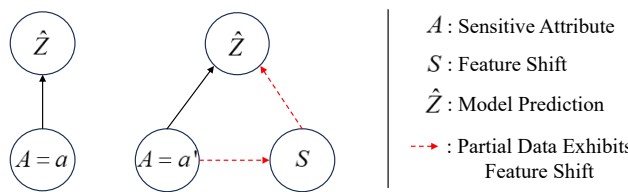

Figure 2: Effect of sensitive attribute and feature shift (FS).

respectively. We assume $A$ and $S$ to be binary, i.e., $A \in \{a, a'\}$ and $S \in \{s, s'\}$, where $s'$ indicates the absence of FS, and it may only occur when $A = a'$, as illustrated in Fig. 2, while our discussion framework can be easily generalized to categorical sensitive attributes and feature shifts. We consider the ID dataset

$$D_{\mathrm{I}} = \{\mathbf{x}_j^{\mathrm{I}}, A_j^{\mathrm{I}}, S_j^{\mathrm{I}}, Z_j^{\mathrm{I}}, Y_j^{\mathrm{I}}\}_{j=1}^n \overset{\text{i.i.d.}}{\sim} P_{\mathrm{I}}, \tag{1}$$

where $P_{\mathrm{I}} = P(X, A, S, Z, Y)$ is the ID joint distribution. And we consider a realistic scenario where both sensitive attributes and FS label variations in OOD data are unobserved, i.e., the OOD dataset is

$$D_{\mathrm{O}} = \{\mathbf{x}_k^{\mathrm{O}}, z_k^{\mathrm{O}}\}_{k=1}^m \overset{\text{i.i.d.}}{\sim} P_{\mathrm{O}}, \tag{2}$$

where $P_{\mathrm{O}} = P(X, Z)$ is the OOD joint distributions. Moreover, we consider the classification model consisting of two components: a feature extractor $\mathbf{f} : \mathcal{X} \to \mathbb{R}^d$, where $\mathbb{R}^d$ is the feature space, and a prediction function $\mathbf{h} : \mathbb{R}^d \to \mathbb{R}^c$ that outputs softmax probabilities.

## 2.2 A REAL-WORLD CASE STUDY

We conduct the case study by employing the BAR (Nam et al., 2020), a real-world action recognition dataset. Additional implementation details are provided in the Appendix E.1.

**For the first question**: Although a model achieves fairness in ID classification, sensitive attributes can still introduce bias in OOD detection, resulting in disproportionately higher error rates among marginalized groups. As shown in Fig. 3(a), the misclassification rate on ID data is identical for all sensitive-attribute values. Nevertheless, the model exhibits disparate OOD detection error rates across sensitive attribute values, revealing clear unfairness.

**For the second question**: FS-induced detection errors are predominantly observed among marginalized groups ($A = a'$). We measured the error rates of the model when the sensitive attribute $A = a'$ while $S$ takes distinct values. As illustrated in Fig. 3(b), across all values of $S$, FS leads to significantly higher error rates in OOD sample detection compared to scenarios without such shift.

## 3 FAIR OOD DETECTION

This section formalizes the problem of fair OOD detection and shows, through counterexamples, that current fairness metrics cannot properly address fairness-confusion issues.

## 3.1 PROBLEM SETUP

To address the challenging issue of incorporating fairness concerns related to sensitive attributes and FS into OOD detection, we need to solve the following critical problem.

**Problem 1** (Fair Out-of-Distribution Detection). *Given a model trained on ID data $D_{\mathrm{I}}$, the objective of fair OOD detection is to enhance both model fairness and OOD detection performance, ensuring that for any test data $\mathbf{x} \in \mathcal{X}$: (a) if $\mathbf{x}$ is an observation from distribution $P_{\mathrm{O}}$, the model can identify $\mathbf{x}$ as OOD data; (b) if $\mathbf{x}$ is an observation from distribution $P_{\mathrm{I}}$, the model can classify $\mathbf{x}$ into its correct ID class; and (c) the prediction is invariant to variations in sensitive attributes or feature shifts.*

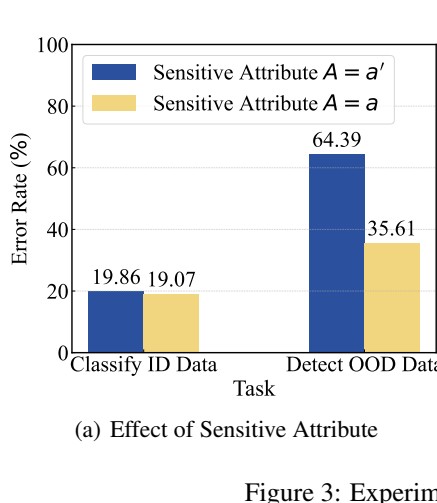 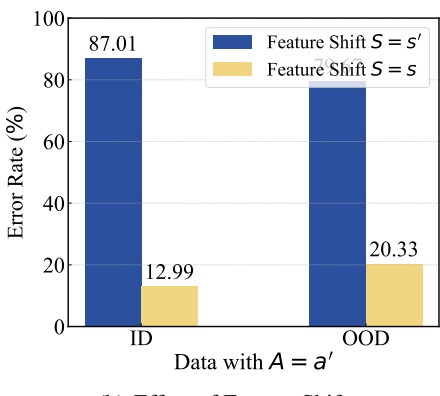

(a) Effect of Sensitive Attribute

(b) Effect of Feature Shift

Figure 3: Experimental results of our case study.

Furthermore, we propose a novel metric, Fair-OOD, that jointly accounts for both sensitive attributes and FS to measure unfairness and FC within a unified framework.

**Definition 1** (Fair-OOD). *In OOD detection, a fair model should simultaneously eliminate both discrimination arising from sensitive attributes A and the effects of the feature shifts S, i.e., the model should satisfy:*

$$P(\widehat{Z} = z | A = a, S = s'_a, Z = z) = P(\widehat{Z} = z | A = a', S = s'_{a'}, Z = z), z \in \mathcal{Z}, \quad (3)$$

*and*

$$P(\widehat{Z} = z | A = a', S = s_{a'}, Z = z) = P(\widehat{Z} = z | A = a', S = s'_{a'}, Z = z), z \in \mathcal{Z}, \quad (4)$$

*where $S = s_a$ and $S = s_{a'}$ represent the values of S when $A = a$ and $A = a'$, respectively.*

**Remark 1.** *Eq. (3) quantifies the effect of sensitive attribute when no FS occurs $(S = s')$, while Eq. (4) measures the impact of FS in the data $(A = a')$ where FS may occur.*

### 3.2 COMPARISON TO OTHER FAIRNESS NOTIONS

In the context of OOD detection, we demonstrate that prior fairness concepts fall short to measure the fairness confusion issue. To illustrate this, we compare with the fairness metric from fair-aware outlier detection because, in existing fairness research, outlier detection is the task most similar to OOD detection. We provide the formal definition of fair-aware outlier detection as follows.

**Definition 2** (Fairness-aware Outlier Detection). *(Shekhar et al., 2021) A model for outlier detection is considered fair if it satisfies both the following fairness metrics:*

- *Demographic Parity (DP):*

$$P(\widehat{Z} = o | A = a) = P(\widehat{Z} = o | A = a'); \quad (5)$$

- *Equal Opportunity (EO):*

$$P(\widehat{Z} = o | A = a, Z = o) = P(\widehat{Z} = o | A = a', Z = o). \quad (6)$$

**Remark 2.** *For consistency in presentation, we retain the variable Z throughout our analysis. It should be noted that in the outlier detection context, $Z = o$ indicates the label of an outlier, while $Z = i$ denotes the label of an inlier.*

Through a counterexample, we demonstrate that conventional fairness metrics (e.g., DP and EO) fail to reliably detect unfairness in OOD detection due to their inability to address fairness confusion induced by feature shifts. Consider the scenario where $P(S = s) = P(S = s') = 0.5$, $P(S = s' | A = a) = 0.99$, and $P(S = s' | A = a') = 0.01$. We construct an unfair model $\mathbf{f}$ with $P(\widehat{Z} = o | A = a, S =$

$s) = 0.1$, $P(\widehat{Z} = o|A = a', S = s') = 0.9$, $P(\widehat{Z} = o|A = a =', S = s) = P(\widehat{Z} = o|A = a, S = s') = 0.5$. Consequently, we obtain that $P(\widehat{Z} = o|A = a) \approx 0.5 = 0.5 \approx P(\widehat{Z} = o|A = a')$ (see Appendix D for the details). Therefore, DP cannot identify unfairness in this scenario. Similarly, EO, being a finer-grained variant of DP, also fails to detect unfairness. This is because these metrics solely consider the effects of the sensitive attribute, overlooking the impact of feature shifts. Distinct from them, our framework explicitly accounts for feature shifts to address the induced fairness confusion.

# 4 PACT: PREDICTIVE ADAPTIVE CALIBRATION

To ensure fairness and consequently achieve more reliable OOD detection, we propose a novel approach, called *Predictive Adaptive Calibration* (PACT), comprises two aspects : **Feature Distribution Regularization** (FDR) and **Predictive Distribution Calibration** (PDC).

## 4.1 FDR: FEATURE DISTRIBUTION REGULARIZATION

For FDR, by constraining the model to extract highly compact feature representations for ID data while maximizing the distance between ID and OOD features, we compel the model to focus exclusively on class-related features while disregarding sensitive attributes and their induced shifted features. Specifically, this is achieved by minimizing:

$$\mathcal{L}_{\text{FDR}} = -\sum_{\mathbf{x}_j^{\text{I}} \in D_{\text{I}}} \log \frac{\psi(\mathbf{x}_j^{\text{I}}, \mathbf{x}^+)}{\psi(\mathbf{x}_j^{\text{I}}, \mathbf{x}^+) + \sum_{\mathbf{x}^- \in D^-} \psi(\mathbf{x}_j^{\text{I}}, \mathbf{x}^-)}, \tag{7}$$

where $\mathbf{x}^+ \in D^+ = \{\mathbf{x}_t^{\text{I}}|y_t = y_j, z_t = z_j, \text{ for } \forall \mathbf{x}_t^{\text{I}} \in D_{\text{I}}\}$ and the $D^- = \{\mathbf{x}_t^{\text{I}}|y_t \neq y_j, z_t = z_j, \text{ for } \forall \mathbf{x}_t^{\text{I}} \in D_{\text{I}}\} \cup D_{\text{O}}$. And $\psi(\cdot, \cdot) = \exp(\text{sim}(\cdot, \cdot)/\tau)$, where $\text{sim}(\cdot, \cdot)$ denotes the cosine similarity, and $\tau$ is the temperature parameter. The following theorem provides guarantees for $\mathcal{L}_{\text{FDR}}$.

**Theorem 1.** *Given a feature extractor* $\mathbf{f}$*, let* $D = \{\mathbf{x}_t\}_{t=1}^N$ *be a set of data points belonging to the same class. For any two non-empty distinct subsets* $D_1, D_2 \subseteq D$ *satisfying* $(1 - \text{sim}(E_{\mathbf{f}}(D_1), E_{\mathbf{f}}(D_2))) \leq \eta$*, where the* $E_{\mathbf{f}}(D_j) = \frac{1}{|D_j|} \sum_{\mathbf{x} \in D_j} \mathbf{f}(\mathbf{x})$ *is the mean feature of the subset* $D_j$ $(j \in \{1, 2\})$*, we have:*

$$V_{\mathbf{f}}(D) \leq \frac{\eta^2}{4}, \tag{8}$$

*where* $V_{\mathbf{f}}(D)$ *denotes feature variance over* $D$.

*Proof.* Due to space limitations, we provide the complete proof in the Appendix C.1. □

**Remark 3.** *Optimizing* $\mathcal{L}_{FDR}$ *effectively minimizes the* $(1 - \text{sim}(E_{\mathbf{f}}(D_1), E_{\mathbf{f}}(D_2)))$*, where both* $D_1$ *and* $D_2$ *contain only a single sample. Theorem 1 demonstrates that optimizing* $\mathcal{L}_{PDR}$ *encourages the model to learn more compact feature representations, ensuring unified feature distributions for data points belonging to the same class but with differing sensitive attributes. This effectively mitigates the influence of sensitive attributes on the learned representations.*

## 4.2 PDC: PREDICTIVE DISTRIBUTION CALIBRATION

To further address FC, we require that the model predictions exhibit robustness to FS in input data, which implies providing identical prediction distributions for both inputs with FS and those without FS. However, in practical scenarios, annotating FS labels for OOD data requires prohibitive effort and is often impractical, thereby preventing targeted fine-tuning using OOD data affected by FS. To this end, our approach mitigates this issue by maximizing the divergence between prediction distributions of ID and OOD data, thus reducing the tendency of the model to produce identical outputs for ID and OOD data with FS. Specifically, we achieve this goal by optimizing the following learning objective.

$$\mathcal{L}_{\text{PDC}} = -\sum_{\mathbf{x}_j^{\text{I}} \in D_{\text{I}}} \sum_{\mathbf{x}_k^{\text{O}} \in D_{\text{O}}} \mathbf{1}\{\hat{y}_j = y_j^{\text{I}}\} \cdot \text{WD}(\mathbf{h}(\mathbf{f}(\mathbf{x}_j^{\text{I}})), \mathbf{h}(\mathbf{f}(\mathbf{x}_k^{\text{O}}))) \tag{9}$$

where $\hat{y}_j = \arg\max_y \mathbf{h}^y(\mathbf{f}(\mathbf{x}_j^{\text{I}}))$, and $\text{WD}(\cdot, \cdot)$ denotes the Wasserstein distance. The following theorem provides theoretical guarantees for $\mathcal{L}_{\text{PDC}}$.

**Theorem 2.** *Given the feature extractor $\mathbf{f}$ and prediction function $\mathbf{h}$, and let $\mathbf{x}' = \mathbf{x} + \delta$ denotes the feature-shifted input. We assume Wasserstein distance satisfies $WD(\mathbf{h} \circ \mathbf{f}(\mathbf{x}), \mathbf{h} \circ \mathbf{f}(\mathbf{x}')) \leq L\|\mathbf{f}(\mathbf{x}) - \mathbf{f}(\mathbf{x}')\|$, where $L$ is the Lipschitz constant of $\mathbf{h}$. When fine-tuning $\mathbf{f}$ and $\mathbf{h}$ by optimizing the loss $\mathcal{L}_{PDC}$, we have: The prediction distribution remains stable under all perturbations $\delta$ with $\|\delta\| \leq \epsilon$ such that:*

$$\|\mathbf{h} \circ \mathbf{f}(\mathbf{x}) - \mathbf{h} \circ \mathbf{f}(\mathbf{x}')\| \leq \epsilon L L_f, \tag{10}$$

*where $L_f$ is the Lipschitz constant of $\mathbf{f}$.*

*Proof.* We also provide the complete proof in the Appendix C.2. $\qquad\square$

**Remark 4.** *Theorem 2 demonstrates that optimizing $\mathcal{L}_{PDC}$ reduces the impact of FS in input data on prediction distributions, thereby mitigating their induced FC.*

### 4.3 OVERALL LEARNING OBJECTIVE

We optimize the classification task using cross-entropy loss $\ell$, i.e., minimizing

$$\mathcal{L}_{\mathrm{ce}} = \frac{1}{n} \sum_{\mathbf{x}_j^{\mathrm{I}}} \ell(\mathbf{h}(\mathbf{f}(\mathbf{x}_j^{\mathrm{I}}), y_j^{\mathrm{I}})). \tag{11}$$

Following Outlier Exposure Hendrycks et al. (2019), we enhance the ability of the model to detect OOD data by minimizing the cross-entropy between model predictions on OOD data and a uniform distribution, i.e.,

$$\mathcal{L}_{\mathrm{oe}} = -\frac{1}{2m} \sum_{\mathbf{x}_k^{\mathrm{O}} \in D_{\mathrm{O}}} \frac{1}{c} \log \mathbf{h}(\mathbf{f}(\mathbf{x}_k^{\mathrm{O}})). \tag{12}$$

Consequently, the overall learning objective is :

$$\mathcal{L} = \mathcal{L}_{\mathrm{ce}} + \mathcal{L}_{\mathrm{oe}} + \alpha \mathcal{L}_{\mathrm{FDR}} + \beta \mathcal{L}_{\mathrm{PDC}}. \tag{13}$$

## 5 EXPERIMENTS

In this section, we conduct extensive experiments to evaluate our proposed metric, Fair-OOD, and method, PACT, by answering the following questions:

- **Q1:** Does Fair-OOD effectively identify *Fairness Confusion* (FC) in reality?
- **Q2:** Does PACT enhance the OOD detection performance of the model?
- **Q3:** Does PACT mitigate bias and FC while preserving high classification accuracy?
- **Q4:** Does debiasing with Fair-OOD conflict with other fairness metrics?
- **Q5:** Do the two key components of PACT perform as anticipated in their intended roles?
- **Q6:** Can our PACT benefit other OOD scoring?

### 5.1 EXPERIMENT SETUP

**Dataset.** We conduct experiments by employ the datasets: `ImageNet-100-C` (Hendrycks & Dietterich, 2018), `BAR` (Nam et al., 2020), and `CIFAR-10-C` (Hendrycks & Dietterich, 2018). In the `BAR`, each class has its own sensitive attributes. `CIFAR-10-C` assigns the same sensitive attribute to all classes. `ImageNet-100-C` represents a more complex and large-scale setting, comprising numerous classes with high-resolution images. Please refer to the Appendix E.3 for more details.

**Baseline.** We compare our proposed method, PACT, with three categories of baseline methods: (1) post-hoc methods in OOD detection, including `MLP` (Hendrycks et al., 2022), `MSP` (Hendrycks & Gimpel, 2017), `T2FNorm` (Regmi et al., 2024a), and `Energy` (Liu et al., 2020); (2) fine-tuning methods in OOD detection with additional auxiliary OOD data in OOD detection, including `OE` (Hendrycks et al., 2019), `Energy-OE` (Liu et al., 2020), and `DAL` (Wang et al., 2023a); and (3) fairness-aware representation learning methods, including `DFA` (Lee et al., 2021), `SelecMix` (Hwang et al., 2022) and `BCSI` (Jung et al., 2024).

**Metric.** We evaluate the OOD detection performance and fairness using the following metrics:

Table 1: Comparison between our proposed PACT and advanced methods across three datasets. For each dataset, we perform a categorical division into ID and OOD partitions. ↑ (or ↓) indicates larger or smaller values are preferred. Results are reported as percentage values, averaged over 10 runs with $\pm_q$ denoting the standard error. Bold font indicates the best results in a column.

| Dataset | Metric | MSP | MLP | T2FNorm | Energy | DFA | SelecMix | BCSI | OE | Energy-OE | DAL | PACT(Ours) |
|---|---|---|---|---|---|---|---|---|---|---|---|---|
| BAR | FPR95 ↓ | 84.53 | 69.54 | 72.07 | 70.94 | 87.44 | 82.13 | 79.01 | 67.15 | 65.91 | 63.48 | **60.99**$_{\pm 0.7}$ |
| | AUROC ↑ | 63.07 | 60.77 | 63.14 | 65.17 | 63.83 | 59.70 | 61.29 | 63.93 | 62.01 | 63.79 | **65.28**$_{\pm 0.5}$ |
| | ACC-U ↑ | 85.42 | 85.42 | 85.67 | 85.42 | 83.01 | 85.67 | 83.29 | 83.59 | 82.37 | 83.02 | **85.94**$_{\pm 0.2}$ |
| | ACC-B ↑ | 50.85 | 50.85 | 54.09 | 50.85 | 56.10 | 57.96 | 57.61 | 63.06 | 62.59 | 65.70 | **77.48**$_{\pm 0.6}$ |
| | DP ↓ | 39.44 | 40.65 | 40.39 | 40.95 | 35.19 | 36.09 | 33.72 | 39.38 | 40.07 | 37.10 | **33.47**$_{\pm 1.7}$ |
| | EO ↓ | 73.72 | 59.33 | 70.92 | 79.32 | 55.10 | 31.20 | 33.72 | 27.69 | 29.77 | 19.77 | **16.67**$_{\pm 0.5}$ |
| | EOD ↓ | 67.03 | 37.01 | 70.50 | 70.21 | 67.59 | 65.60 | 60.59 | 30.91 | 31.17 | 28.04 | **25.16**$_{\pm 1.2}$ |
| | Fair-OOD ↓ | 27.10 | 17.97 | 17.81 | 18.72 | 15.80 | 15.90 | 15.07 | 16.67 | 16.77 | 15.95 | **4.17**$_{\pm 0.0}$ |
| CIFAR-10-C | FPR95 ↓ | 88.02 | 84.55 | 82.76 | 87.44 | 87.14 | 85.17 | 81.19 | 51.92 | 50.70 | 46.52 | **38.79**$_{\pm 0.3}$ |
| | AUROC ↑ | 60.57 | 63.04 | 64.29 | 61.41 | 62.49 | 60.07 | 65.97 | 86.23 | 87.79 | 88.91 | **89.50**$_{\pm 0.2}$ |
| | ACC-U ↑ | 81.29 | 81.29 | 81.31 | 81.29 | 81.22 | 81.51 | 81.09 | 80.75 | 79.10 | 81.10 | **81.55**$_{\pm 0.2}$ |
| | ACC-B ↑ | 69.11 | 69.11 | 69.09 | 69.11 | 68.70 | 68.76 | 70.18 | 72.10 | 72.96 | 75.51 | **78.27**$_{\pm 0.3}$ |
| | DP ↓ | 9.34 | 7.86 | 5.28 | 9.42 | 4.73 | 5.71 | 3.97 | 3.58 | 3.56 | 3.29 | **0.50**$_{\pm 0.0}$ |
| | EO ↓ | 12.90 | 5.06 | 4.19 | 5.61 | 2.95 | 2.67 | 2.62 | 3.39 | 3.79 | 2.92 | **2.48**$_{\pm 0.1}$ |
| | EOD ↓ | 7.74 | 6.69 | 4.06 | 7.83 | 3.94 | 5.17 | 5.65 | 6.21 | 6.70 | 6.12 | **1.30**$_{\pm 0.2}$ |
| | Fair-OOD ↓ | 6.55 | 5.63 | 9.88 | 5.27 | 5.00 | 5.92 | 6.77 | 4.91 | 5.72 | 5.21 | **1.59**$_{\pm 0.1}$ |
| ImageNet-100-C | FPR95 ↓ | 61.16 | 56.81 | 66.03 | 56.63 | 87.43 | 85.77 | 81.70 | 35.54 | 37.21 | 29.97 | **17.50**$_{\pm 0.3}$ |
| | AUROC ↑ | 76.80 | 78.71 | 75.25 | 78.33 | 61.39 | 62.70 | 65.17 | 91.79 | 89.03 | 92.26 | **96.43**$_{\pm 0.2}$ |
| | ACC-U ↑ | 81.89 | 81.89 | 81.70 | 81.89 | 81.22 | 83.31 | 83.93 | 84.78 | 83.07 | 83.29 | **87.41**$_{\pm 0.5}$ |
| | ACC-B ↑ | 62.96 | 62.96 | 61.59 | 62.96 | 63.71 | 65.92 | 65.70 | 67.18 | 66.50 | 67.96 | **74.03**$_{\pm 0.3}$ |
| | DP ↓ | 24.51 | 25.72 | 25.10 | 25.99 | 7.84 | 7.03 | 6.92 | 14.70 | 15.97 | 15.05 | **6.69**$_{\pm 0.2}$ |
| | EO ↓ | 5.27 | 4.02 | 5.92 | 3.88 | 6.97 | 5.25 | 5.09 | 4.50 | 5.67 | 5.01 | **3.96**$_{\pm 0.2}$ |
| | EOD ↓ | 13.63 | 19.84 | 18.5 | 19.99 | 13.39 | 12.12 | 12.60 | 12.83 | 13.97 | 13.01 | **8.43**$_{\pm 0.7}$ |
| | Fair-OOD ↓ | 15.28 | 15.50 | 17.75 | 17.50 | 15.24 | 15.90 | 13.71 | 13.24 | 12.97 | 13.59 | **9.56**$_{\pm 0.2}$ |

- For OOD detection: the false positive rate at which OOD samples are declared as ID when 95% of ID samples are declared as ID (FPR95); and the area under the receiver operating characteristic curve (AUROC);

- For ID classification: the classification accuracy on ID data with sensitive attribute $A = a$ (ACC-U) and on ID data with $A = a'$ (ACC-B);

- For Fairness: Demographic Parity (DP) computed by DP $= |P(\widehat{Z} = o|A = a) - P(\widehat{Z} = o|A = a')|$; Equal Opportunity (EO) computed by EO $= |P(\widehat{Z} = o|A = a, Z = o) - P(\widehat{Z} = o|A = a', Z = o)|$; and Equalized Odds (EOD) computed by EOD $= \frac{1}{2}|P(\widehat{Z} = o|A = a, Z = o) - P(\widehat{Z} = o|A = a', Z = o)| + |P(\widehat{Z} = o|A = a, Z = i) - P(\widehat{Z} = o|A = a', Z = i)|$; and our Fair-OOD computed by

$$\text{Fair-OOD} = \frac{1}{4} \sum_{z \in \{i,o\}} \Big\{ |P(\widehat{Z} = z|A = a, S = s'_a, Z = z) - P(\widehat{Z} = z|A = a', S = s'_{a'}, Z = z)|$$

$$+ |P(\widehat{Z} = z|A = a', S = s_{a'}, Z = z) - P(\widehat{Z} = z|A = a', S = s'_{a'}, Z = z)| \Big\}.$$

where the values of $S$ are determined by the same scheme in our case study.

**PACT Default Setup.** We employ ResNet-18 (He et al., 2016) as the backbone model and fine-tune it using stochastic gradient descent with Nesterov momentum (Duchi et al., 2011). The weight decay coefficient is set to 0.0005, the momentum to 0.09, and the learning rate to 0.0001. We fine-tune for 3 epochs with batch sizes of 128 for both ID and OOD data. We set $\tau = 0.1$, $\beta = 2$, and $\gamma = 2$. Please refer to the Appendix E.2 for more details.

## 5.2 MAIN RESULT

We primally highlight the following observations:

(a) **Fair-OOD captures the FC in OOD detection (Q1).** As shown in Table 1 and Fig. 4(a), prior to applying PACT, Fair-OOD remain significantly higher across all three datasets. For instance, the

`MSP` yields a Fair-OOD score of 27.10% on the `BAR` dataset. This observation aligns with our case study findings that OOD detection is not only influenced by sensitive attributes but also exhibits FC.

(b) **PACT significantly improves OOD detection performance (Q2).** Across all three datasets, our proposed PACT framework demonstrates significant improvements in OOD detection performance. Most notably, on the large-scale, multi-class ImageNet-100-C benchmark, our PACT achieves a substantial 12.47% absolute improvement over the strongest baseline. In contrast, existing debiasing methods show negligible performance gains in OOD detection, exhibiting consistently weak detection capabilities. These results validate reliable performance of our PACT.

(c) **PACT not only mitigates bias and resolves fairness confusion, but also improves classification accuracy (Q3).** As evidenced in Table 1, conventional debiasing methods demonstrate competent performance on standard fairness metrics but consistently fail to optimize Fair-OOD. In contrast, our PACT simultaneously optimizes both conventional fairness metrics and Fair-OOD, effectively mitigating FC. This also demonstrates that **debiasing with Fair-OOD is not in conflict with conventional fairness metrics (Q4).** Furthermore, while existing debiasing methods and prior OOD detection approaches exhibit significant disparities in classification accuracy across different sensitive attribute values, PACT-optimized models achieve substantially closer ACC-U and ACC-B values.

### 5.3 IN-DEPTH ANALYSIS

**The two key components of PACT, FDR and PDC, function as designed in their specified roles (Q5).** From Table 2, we can observe that: On one hand, compared to not using PACT, employing only the FDR optimization improves the average accuracy from 72.42% to 79.07%. This enhancement stems from the increased accuracy of ACC-B, ensuring comparable performance of the model across different sensitive attributes. This

Table 2: Ablation study on key components of PACT.

| Index | Component | | Metric | | |
|-------|-----------|-----------|-------------|-----------|--------|
| | $\mathcal{L}_{\text{FDR}}$ | $\mathcal{L}_{\text{PDC}}$ | Average ACC↑ | Fair-OOD↓ | FPR95↓ |
| 1 | | | 72.42 | 25.00 | 65.98 |
| 2 | ✓ | | 79.07 | 16.67 | 64.44 |
| 3 | | ✓ | 72.68 | 8.33 | 61.57 |
| 4 | ✓ | ✓ | **81.71** | **4.17** | **60.99** |

result demonstrates the debiasing capability of FDR. On the other hand, the Fair-OOD metric, optimized by PDC, decreases from 25% to 8.33%, validating its effectiveness in handling FC. Furthermore, the combined use of FDR and PDC achieves the best overall performance. These findings collectively substantiate the efficacy of the individual components within PACT.

**Employing PACT to optimize models can benefit other OOD scoring methods (Q6).** From Figure 4(a), PACT optimization significantly reduces Fair-OOD metric values across all OOD scoring methods. This observation substantiates both the effectiveness and general applicability of PACT across different OOD detection approaches.

**Parameter Analysis.** As shown in Fig. 4(b), PACT consistently outperforms the state-of-the-art (SOTA) method across a broad range of parameter configurations: $\beta \in [0, 6]$, $\tau \in [0.1, 0.5]$, and all values of $\alpha$. This demonstrates that PACT maintains robust performance without requiring meticulous fine-tuning, indicating its strong generalization capability across diverse hyperparameter settings.

## 6 RELATED WORK

In this section, we review the research in fairness-aware representation learning and OOD detection.

**Fairness-aware Representation Learning** The existing fairness-aware representation learning methods can be primarily categorized into the following approaches: reweighting, data augmentation, feature disentanglement, robust optimization, along with other miscellaneous techniques that do not fall into these main categories. Sample reweighting represents one of the earliest proposed algorithmic approaches in this domain, which mitigates model overfitting to biased attributes in majority groups by assigning higher weights to minority group instances compared to majority group samples (Roh et al., 2023; Cheng et al., 2024; Kim et al., 2022). Data augmentation methods enhance model fairness by improving generalization capabilities across diverse data populations through strategically expanded training set diversity (Kim et al., 2021; Ramaswamy et al., 2021; Li et al.,

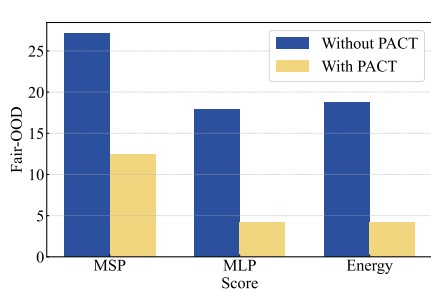
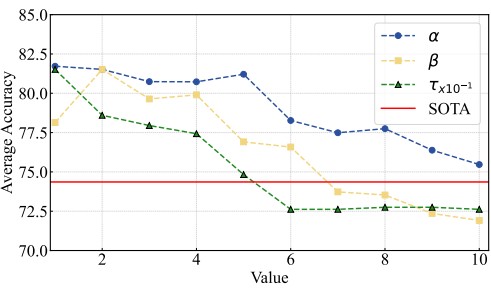

(a) Different OOD Scores    (b) Hyper-parameter Sensitivity Analysis

Figure 4: Fig. 4(a) presents Fair-OOD results obtained with different OOD scores before and after training with PACT. Fig. 4(b) illustrates the sensitivity analysis of hyper-parameters $\alpha$, $\beta$, and $\tau$.

2025a). Feature disentanglement methods aim to decouple the correlation between target features and bias attributes in the latent representation space, thereby enabling models to learn bias-invariant target features and mitigate interference from spurious biased correlations (Ragonesi et al., 2021; Zhu et al., 2021; Zhang et al., 2025). Robust optimization enhances model fairness by minimizing the worst-case loss over diverse training-set subsets or formally defined perturbation sets surrounding the data distribution (Mandal et al., 2020; Du & Wu, 2021; Li et al., 2025b). In addition, other methods improve model fairness through architectural modifications (Shrestha et al., 2022; Sreelatha et al., 2024) or post-processing techniques applied to the model outputs (Basu et al., 2024). However, existing fairness-aware representation learning methods exhibit critical limitations in OOD detection task, particularly in addressing potential fairness confusion issue.

### 6.1 OUT-OF-DISTRIBUTION DETECTION

**Out-of-Distribution Detection** Current OOD detection methods can be broadly categorized into three approaches: post-hoc methods, representation-based methods, and the outlier exposure. Post-hoc methods focus on designing various OOD scoring functions to distinguish between ID and OOD data patterns. These approaches leverage different types of information from model outputs. For instance, researchers (Hendrycks & Gimpel, 2017; Liu et al., 2020; Peng et al., 2024) develop OOD scoring functions using the logit outputs of model, while other methods utilize extracted features (Wang et al., 2022a; Regmi et al., 2024b; Luo et al., 2023) or gradient information (Liang et al., 2018; Huang et al., 2021; Igoe et al., 2022) for this purpose. Representation-based methods posit that discriminative ID representations are pivotal for OOD. They enhance the quality of these representations via contrastive learning (Sehwag et al., 2021; Wang et al., 2022b), data augmentation (Tack et al., 2020), or regularization imposed on embedding features and model outputs (Du et al., 2022; Ming et al., 2023; Zhang et al., 2024; Wei et al., 2022). Outlier exposure augments model training with auxiliary OOD data, thereby equipping the classifier with explicit OOD knowledge and sharpening its discriminative boundary between ID and OOD instances (Hendrycks et al., 2019; Wang et al., 2023a;b; 2025; 2023c). However, current OOD detection methods neglect fairness considerations, potentially degrading both detection performance and model reliability.

### 7 CONCLUSION

In this paper, we incorporate fairness considerations into OOD detection and introduce a challenging problem, *Fair OOD Detection*, which involves biases induced by sensitive attributes and *Fairness Confusion* (FC) arising from *Feature Shifts* (FS) triggered by these attributes. To quantify FC, we propose a novel metric, Fair-OOD. Furthermore, we propose *Predictive Adaptive Calibration* (PACT), a method that jointly enhances OOD detection performance, mitigates bias, and alleviates FC. Extensive experiments validate the effectiveness of Fair-OOD in identifying FC and the efficacy of PACT in delivering robust and fairness-aware OOD detection. We hope our work will inspire future research on OOD detection and fairness-aware representation learning.

## 8 ETHICS STATEMENT

This study complies with the ICLR Code of Ethics. We propose a novel OOD detection framework and evaluate it on publicly available benchmark datasets. These datasets contain no personally identifiable or sensitive information, thereby ensuring no risks to privacy or security. Our research advances the application of OOD detection in more practical scenarios and holds potential scientific and technological value. All experimental protocols are transparently documented and fairly compared with prior work. The contributions of this study are intended solely for research, supporting the development of artificial intelligence.

## 9 REPRODUCIBILITY STATEMENT

We provide detailed descriptions of our framework, theoretical results, and experimental settings in the paper and appendix. All datasets used are publicly available, and the current description of our method is sufficient for full reproducibility. If the paper is accepted, we will be glad to release the complete implementation to further support the research community.

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

SUPPLEMENTARY MATERIAL: FAIR OUT-OF-DISTRIBUTION DETECTION

## A  NOTATIONS

In this section, we summarize the key notations in Table 3.

Table 3: Notations and associated descriptions.

| | Description |
|---|---|
| | **Variable and Space** |
| $\mathcal{X}$ and $\mathcal{Y}$ | feature and ID label spaces |
| $X^{\mathrm{I}}$, $X^{\mathrm{O}}$, and $Y^{\mathrm{I}}$ | ID and OOD feature variables, and ID label variable |
| $Z$ and $\mathcal{Z}$ | ID/OOD binary variable and space |
| $A$ and $S$ | sensitive attribute and feature shift |
| | **Distribution and Measurement** |
| $P_{\mathrm{I}}$ and $P_{\mathrm{O}}$ | ID and OOD joint distributions |
| $\phi(\cdot, \cdot)$ | distance metric |
| $\mathrm{sim}(\cdot, \cdot)$ | cosine similarity |
| | **Data and Model** |
| $D_{\mathrm{I}}$ and $D_{\mathrm{O}}$ | ID and OOD datasets |
| $n$ and $m$ | numbers of ID data and OOD data |
| $\mathbf{x}$, $y$, and $z$ | instance, ID label, and ID/OOD binary label |
| $\mathbf{f}(\cdot)$ and $\mathbf{h}(\cdot)$ | feature extractor and prediction function |
| | **Loss and Risk** |
| $\ell$ | cross entropy loss |
| $\mathcal{L}_{\mathrm{ce}}$ | multi-class classification |
| $\mathcal{L}_{\mathrm{oe}}$ | ID/OOD classification |
| | **PACT** |
| $\alpha$, $\beta$, and $\tau$ | trade-off parameters, and temperature parameter |
| $\mathcal{L}_{\mathrm{FDR}}$ | feature distribution regularization |
| $\mathcal{L}_{\mathrm{PDC}}$ | predictive distribution calibration |

## B  ALGORITHM: PACT

The overall algorithm of our proposed PACT is presented in Algorithm 1. During each training epoch, a minibatch is randomly sampled from the training data, and the model is optimized through a composite objective function that integrates several key components. First, the feature distribution regularization term, which mitigates sensitive attribute biases, is computed using Eq. (7), while the predictive distribution calibration term, which addresses feature shift, is computed using Eq. (9). Furthermore, Eq. (11) optimizes multi-class classification on ID data via cross-entropy loss, while Eq. (12) performs binary classification to separate ID and OOD samples. These components are aggregated into a unified training objective in Eq. (13), enabling end-to-end optimization.

## C  THEORETICAL GUARANTEES OF PACT

In this section, we present complete theoretical proofs for Theorems 1 and 2 stated in the main body.

### C.1  THEOREM 1 WITH COMPLETE PROOF

To facilitate readability, we restate Theorem 1 here.

**Theorem 1.** *Given a feature extractor* $\mathbf{f}$, *let* $D = \{\mathbf{x}_t\}_{t=1}^{N}$ *be a set of data points belonging to the same class. For any two non-empty distinct subsets* $D_1, D_2 \subseteq D$ *satisfying* $(1 - \mathrm{sim}(E_{\mathbf{f}}(D_1), E_{\mathbf{f}}(D_2))) \leq \eta$, *where the* $E_{\mathbf{f}}(D_j) = \frac{1}{|D_j|} \sum_{\mathbf{x} \in D_j} \mathbf{f}(\mathbf{x})$ *is the mean feature of the*

---

**Algorithm 1** Predictive Adaptive Calibration (PACT)

---

**Input**: The ID training dataset $D_I$; the training OOD dataset $D_O$; the pre-trained model with extractor $\mathbf{f}$ and prediction function $\mathbf{h}$.
**Parameter**: The trade-off parameters $\alpha$ and $\beta$; the temperature parameter $\tau$.
**Output**: The well-trained model.

1: # Training:
2: **for** $epoch \leftarrow 1$ **to** $epochs$ **do**
3:     Fetch a mini-batch $(\mathcal{B}_a^I, \mathcal{B}_{a'}^I, \mathcal{B}^O)$;
4:     Compute the feature distribution regularization term $\mathcal{L}_{FDR}$ in Eq. (7);
5:     Compute the predictive distribution calibration term $\mathcal{L}_{PDC}$ in Eq. (9);
6:     Compute $\mathcal{L}_{ce}$ and $\mathcal{L}_{oe}$ in Eqs. (11) and (12);
7:     Train the model by minimizing $\mathcal{L}$ in Eq. (13));
8: **end for**
9: **return** The well-trained model.

---

*subset $D_j$ $(j \in \{1, 2\})$, we have:*

$$V_{\mathbf{f}}(D) \leq \frac{\eta^2}{4}, \tag{14}$$

*where $V_{\mathbf{f}}(D)$ denotes feature variance over $D$.*

*Proof.* Let $\mu$ denote the global mean, i.e.,

$$\mu = \frac{1}{N} \sum_{t=1}^{N} \mathbf{f}(\mathbf{x}_t).$$

Consider singleton subsets $D_1 = \{\mathbf{x}_k\}$ and $D_2 = \{\mathbf{x}_l\}$, where $\mathbf{x}_k, \mathbf{x}_l$ are arbitrary samples from the same category. According to $(1 - \text{sim}(E_{\mathbf{f}}(D_1), E_{\mathbf{f}}(D_2))) \leq \eta$, we have:

$$\xi(\mathbf{x}_k, \mathbf{x}_l) = (1 - \text{sim}(E_{\mathbf{f}}(D_1), E_{\mathbf{f}}(D_2))) \leq \eta,$$

where $\xi(\cdot, \cdot)$ is the cosine distance. Thus, the above inequality demonstrates that the feature distance between any two samples does not exceed $\eta$. This implies all $\{\mathbf{x}_t\}_{t=1}^{N}$ lie within a closed ball.

Furthermore, since the centroid is a convex combination of points, it must reside within this ball. Consequently, the above conclusion extends to any non-empty, non-singleton subset.

Then, for any $\mathbf{x}_k$ and $\mathbf{x}_l$, we have:

$$\xi(\mathbf{f}(\mathbf{x}_k) - \mathbf{f}(\mathbf{x}_l)) \leq \xi(\mathbf{f}(\mathbf{x}_k) - \mu) + \xi(\mathbf{f}(\mathbf{x}_l) - \mu)$$
$$\leq 2\max\{\xi(\mathbf{f}(\mathbf{x}_k) - \mu), \xi(\mathbf{f}(\mathbf{x}_l) - \mu)\}.$$

According to $\xi(\mathbf{f}(\mathbf{x}_k) - \mathbf{f}(\mathbf{x}_l)) \leq \eta$, we have:

$$2\max\{\xi(\mathbf{f}(\mathbf{x}_k) - \mu), \xi(\mathbf{f}(\mathbf{x}_l) - \mu)\} \leq \eta$$
$$\max\{\xi(\mathbf{f}(\mathbf{x}_k) - \mu), \xi(\mathbf{f}(\mathbf{x}_l) - \mu)\} \leq \frac{\eta}{2}.$$

Since $\mathbf{x}_k$ and $\mathbf{x}_l$ are arbitrary, it follows that:

$$\xi(\mathbf{f}(\mathbf{x}_t), \mu) \leq \frac{\eta}{2}, \forall \mathbf{x}_t \in D.$$

Then, we have:

$$V_{\mathbf{f}}(D) = \frac{1}{N} \sum_{t=1}^{N} \xi(\mathbf{f}(\mathbf{x}_t), \mu)^2 \leq \frac{1}{N} \sum_{t=1}^{N} (\frac{\eta}{2})^2 \leq \frac{\eta^2}{4}.$$

Thus, the proof is completed.

$\square$

## C.2 THEOREM 2 WITH COMPLETE PROOF

We also restate Theorem 2 here.

**Theorem 2.** *Given the feature extractor* $\mathbf{f}$ *and prediction function* $\mathbf{h}$, *and let* $\mathbf{x}' = \mathbf{x} + \delta$ *denotes the feature-shifted input. We assume Wasserstein distance satisfies* $\mathtt{WD}(\mathbf{h} \circ \mathbf{f}(\mathbf{x}), \mathbf{h} \circ \mathbf{f}(\mathbf{x}')) \leq L\|\mathbf{f}(\mathbf{x}) - \mathbf{f}(\mathbf{x}')\|$, *where* $L$ *is the Lipschitz constant of* $\mathbf{h}$. *When fine-tuning* $\mathbf{f}$ *and* $\mathbf{h}$ *by optimizing the loss* $\mathcal{L}_{PDC}$, *we have: The prediction distribution remains stable under all perturbations* $\delta$ *with* $\|\delta\| \leq \epsilon$ *such that:*

$$\|\mathbf{h} \circ \mathbf{f}(\mathbf{x}) - \mathbf{h} \circ \mathbf{f}(\mathbf{x}')\| \leq \epsilon L L_f, \tag{15}$$

*where* $L_f$ *is the Lipschitz constant of* $\mathbf{f}$.

*Proof.* For a finite-dimensional neural network with weight matrix $W$, all singular values of $W$ are finite, and thus its spectral norm is bounded. Since the Lipschitz constant $L$ of the network is given by the spectral norm (Miyato et al., 2018), it follows that $L$ is also bounded. Therefore, the Lipschitz constant is explicitly constrained. According to $\mathtt{WD}(\mathbf{h} \circ \mathbf{f}(\mathbf{x}), \mathbf{h} \circ \mathbf{f}(\mathbf{x}')) \leq L\|\mathbf{f}(\mathbf{x}) - \mathbf{f}(\mathbf{x}')\|$, as $\mathtt{WD}(\mathbf{h} \circ \mathbf{f}(\mathbf{x}), \mathbf{h} \circ \mathbf{f}(\mathbf{x}'))$ increases, it asymptotically approaches its upper bound $L\|\mathbf{f}(\mathbf{x}) - \mathbf{f}(\mathbf{x}')\|$.

Given that $\mathbf{x}' = \mathbf{x} + \delta$, where $\|\delta\| \leq \epsilon$, by applying the Lipschitz condition, we obtain:

$$\|\mathbf{f}(\mathbf{x}) - \mathbf{f}(\mathbf{x}')\| \leq L_f \|\mathbf{x} - \mathbf{x}'\| = L_f \|\delta\| \leq L_f \epsilon.$$

Then, we have:

$$\mathtt{WD}(\mathbf{h} \circ \mathbf{f}(\mathbf{x}), \mathbf{h} \circ \mathbf{f}(\mathbf{x}')) \leq L\|\mathbf{f}(\mathbf{x}) - \mathbf{f}(\mathbf{x}')\| \leq \epsilon L L_f.$$

Thus, the proof is completed.

$\square$

# D DETAILS ON COMPARISON BETWEEN OUR FAIR-OOD AND OTHER FAIRNESS NOTION

To formally distinguish between Fair-OOD and DP, we construct a counterexample. To make it more intuitive, we present the unfair decision model in Table 4, along with $P(S = s) = P(S = s') = 0.5$, $P(S = s'|A = a) = 0.99$, and $P(S = s'|A = a') = 0.01$. Then, we derive the DP as follows.

Table 4: Unfair desision model.

| $A$ | $S$ | $P(\widehat{Z} = o|A, S)$ |
|---|---|---|
| $a$ | $s$ | 0.1 |
| $a$ | $s'$ | 0.5 |
| $a'$ | $s$ | 0.5 |
| $a'$ | $s'$ | 0.9 |

$P(\widehat{Z} = o|A = a)$

$= P(\widehat{Z} = o|S = s, A = a)P(S = s|A = a) + P(\widehat{Z} = o|S = s', A = a)P(S = s'|A = a)$

$= 0.001 + 0.495 = 0.496 \approx 0.5.$

$P(\widehat{Z} = o|A = a')$

$= P(\widehat{Z} = o|S = s, A = a')P(S = s|A = a') + P(\widehat{Z} = o|S = s', A = a')P(S = s'|A = a')$

$= 0.495 + 0.009 = 0.504 \approx 0.5.$

Consequently, we obtain that $P(\widehat{Z} = o|A = a) \approx 0.5 = 0.5 \approx P(\widehat{Z} = o|A = a')$. Therefore, DP cannot detect unfairness in this scenario, where the bias can also be induced by feature shift, but our Fair-OOD can.

# E ADDITIONAL EXPERIMENTAL DETAILS

In this section, we present details for the empirical study and experiments on the main body.

Table 5: Sensitive attributes for each class in BAR.

| Class | Sensitive Attribute | |
|---|---|---|
| | $A = a$ | $A = a'$ |
| Climbing | Rock Wall | |
| Diving | Underwater | |
| Fishing | Water Surface | Other Background |
| Pole Vaulting | Sky | |
| Racing | A Paved Track | |
| Throwing | Playing Field | |

### E.1 ADDITIONAL DETAILS OF EMPIRICAL STUDY

Our case study is conducted by employing ResNet-18 as the model. We train the model to ensure fair classification performance on ID data. The distance metric is the Euclidean distance.

Furthermore, we determine the values of $S$ using the following approach.

$$S(\mathbf{x}_t) = \begin{cases} s, & \text{if } \min_{y \in \mathcal{Y}} \phi(\mathbf{x}_t, \mu^y) < \tau, \\ s', & \text{if, } \min_{y \in \mathcal{Y}} \phi(\mathbf{x}_t, \mu^y) \geq \tau, \end{cases} \tag{16}$$

where $\phi(\cdot, \cdot)$ is a distance metric, $\mu^y$ represents the feature centroid corresponding to the $y$-th class, and we compute it by $\mu^y = \frac{1}{n^y} \sum_{i=1}^{n^y} \mathbf{x}_i^y$, where $n^y$ denotes the number of samples in the $y$-th class. The threshold $\tau$ is set as the 70th percentile of the ordered set of ID sample-centroid distances $\phi(\mathbf{x}, \mu^y)$, namely,

$$\tau = \text{SORT}_{70}\{\phi(\mathbf{x}_i^y, \mu^y); \text{ for } \mathbf{x}_i^y \in D_{\text{I}}\}, \tag{17}$$

where $\text{SORT}_{70}\{\cdot\}$ sorts a set to its increasing order and return the 70th percentile element from the results, and the 70th percentile is selected as the threshold because feature-shifted samples typically exhibit larger distances to the original class centroids.

### E.2 ADDITIONAL DETAILS OF EXPERIMENTS

**Pre-training Setups.** We employ ResNet-18 (He et al., 2016) as the backbone network. We pre-training the model for 50 epochs with cross-entropy loss. The optimizer is SGD with a momentum of 0.9, an initial learning rate of 0.01, and a cosine annealing decay schedule (Loshchilov & Hutter, 2017). Additionally, a weight decay of 0.0005 is applied.

**Details of Implementation.** During expanded experimental investigations, we fine-tune the model for 3 epochs on BAR and CIFAR-10-C, and for 10 epochs on ImageNet-100-C. We use MLP scores as the OOD score of our method PACT during the testing phase. We retain the default hyperparameter settings for all baseline methods.

### E.3 DATASETS

**BAR** (Nam et al., 2020) is a real-world action recognition dataset. We summarize the sensitive attributes for each class in Table 5. We designate data belonging to four classes as ID data and utilize data from the remaining two classes as OOD data.

**CIFAR-10-C** (Hendrycks & Dietterich, 2018) is derived from CIFAR-10 (Krizhevsky & Hinton, 2009) by applying various corruptions, including gaussian noise, defocus blur, glass blur, impulse noise, shot noise, snow, and zoom blur. We use CIFAR-10 as the data with $A = a$, and CIFAR-10-C as the data with $A = a'$. We designate data belonging to seven classes as ID data and utilize data from the remaining three classes as OOD data.

**ImageNet-100-C** (Hendrycks & Dietterich, 2018) is derived from ImageNet-100 (Deng et al., 2009), which is the subset of ImageNet-1k, containing 100 distinct classes, and consists of multiple types of algorithmically generated corruptions from noise, blur, weather, and digital categories. We use ImageNet-100 as the data with $A = a$, and ImageNet-100-C as the data with $A = a'$. We

designate data belonging to 70 classes as ID data and utilize data from the remaining 30 classes as OOD data. We present class labels along with their corresponding WordNet IDs below.

n01968897 n01770081 n01818515 n02011460 n01496331 n01847000 n01687978 n01740131
n01537544 n01491361 n02007558 n01735189 n01630670 n01440764 n01819313 n02002556
n01667778 n01755581 n01924916 n01751748 n01984695 n01729977 n01614925 n01608432
n01443537 n01770393 n01855672 n01560419 n01592084 n01914609 n01582220 n01667114
n01985128 n01820546 n01773797 n02006656 n01986214 n01484850 n01749939 n01828970
n02018795 n01695060 n01729322 n01677366 n01734418 n01843383 n01806143 n01773549
n01775062 n01728572 n01601694 n01978287 n01930112 n01739381 n01883070 n01774384
n02037110 n01795545 n02027492 n01531178 n01944390 n01494475 n01632458 n01698640
n01675722 n01877812 n01622779 n01910747 n01860187 n01796340 n01833805 n01685808
n01756291 n01514859 n01753488 n02058221 n01632777 n01644900 n02018207 n01664065
n02028035 n02012849 n01776313 n02077923 n01774750 n01742172 n01943899 n01798484
n02051845 n01824575 n02013706 n01955084 n01773157 n01665541 n01498041 n01978455
n01693334 n01950731 n01829413 n01514668

"chambered nautilus, pearly nautilus, nautilus" "harvestman, daddy longlegs, Phalangium opilio" "macaw" "bittern" "electric ray, crampfish, numbfish, torpedo" "drake" "agama" "night snake, Hypsiglena torquata" "indigo bunting, indigo finch, indigo bird, Passerina cyanea" "tiger shark, Galeocerdo cuvieri" "flamingo" "garter snake, grass snake" "common newt, Triturus vulgaris" "tench, Tinca tinca" "sulphur-crested cockatoo, Kakatoe galerita, Cacatua galerita" "white stork, Ciconia ciconia" "terrapin" "diamondback, diamondback rattlesnake, Crotalus adamanteus" "flatworm, platyhelminth" "sea snake" "spiny lobster, langouste, rock lobster, crawfish, crayfish, sea crawfish" "green snake, grass snake" "bald eagle, American eagle, Haliaeetus leucocephalus" "kite" "goldfish, Carassius auratus" "scorpion" "goose" "bulbul" "chickadee" "sea anemone, anemone" "magpie" "mud turtle" "crayfish, crawfish, crawdad, crawdaddy" "lorikeet" "garden spider, Aranea diademata" "spoonbill" "hermit crab" "great white shark, white shark, man-eater, man-eating shark, Carcharodon carcharias" "green mamba" "bee eater" "bustard" "Komodo dragon, Komodo lizard, dragon lizard, giant lizard, Varanus komodoensis" "hognose snake, puff adder, sand viper" "common iguana, iguana, Iguana iguana" "king snake, kingsnake" "toucan" "peacock" "barn spider, Araneus cavaticus" "wolf spider, hunting spider" "thunder snake, worm snake, Carphophis amoenus" "water ouzel, dipper" "Dungeness crab, Cancer magister" "nematode, nematode worm, roundworm" "vine snake" "wombat" "black widow, Latrodectus mactans" "oystercatcher, oyster catcher" "black grouse" "red-backed sandpiper, dunlin, Erolia alpina" "goldfinch, Carduelis carduelis" "snail" "hammerhead, hammerhead shark" "spotted salamander, Ambystoma maculatum" "American alligator, Alligator mississipiensis" "banded gecko" "wallaby, brush kangaroo" "great grey owl, great gray owl, Strix nebulosa" "jellyfish" "black swan, Cygnus atratus" "ptarmigan" "hummingbird" "whiptail, whiptail lizard" "sidewinder, horned rattlesnake, Crotalus cerastes" "hen" "horned viper, cerastes, sand viper, horned asp, Cerastes cornutus" "albatross, mollymawk" "axolotl, mud puppy, Ambystoma mexicanum" "tailed frog, bell toad, ribbed toad, tailed toad, Ascaphus trui" "American coot, marsh hen, mud hen, water hen, Fulica americana" "loggerhead, loggerhead turtle, Caretta caretta" "redshank, Tringa totanus" "crane" "tick" "sea lion" "tarantula" "boa constrictor, Constrictor constrictor" "conch" "prairie chicken, prairie grouse, prairie fowl" "pelican" "coucal" "limpkin, Aramus pictus" "chiton, coat-of-mail shell, sea cradle, polyplacophore" "black and gold garden spider, Argiope aurantia" "leatherback turtle, leatherback, leathery turtle, Dermochelys coriacea" "stingray" "rock crab, Cancer irroratus" "green lizard, Lacerta viridis" "sea slug, nudibranch" "hornbill" "cock"

# F THE USE OF LARGE LANGUAGE MODELS (LLMS)

The authors declare that large language models (LLMs) were used solely for polishing the writing of this manuscript. No part of the theoretical development, algorithm design, experimental implementation, data analysis, or other research-related tasks involved the use of LLMs.

