# OpenReview forum: "Fair Out-of-Distribution Detection"
_ICLR.cc/2026/Conference — Submitted to ICLR 2026_

### Official Review · Reviewer_jx7G · 2025-10-20

**Soundness:** 2
**Presentation:** 3
**Contribution:** 2
**Rating:** 4
**Confidence:** 4

**Summary:**

The paper proposes predictive adaptive calibration, a method that combines feature distribution regularization and predictive distribution calibration to simultaneously improve OOD detection, mitigate "fairness" confusion caused by "sensitive" attribute.

**Strengths:**

- The proposed PACT framework includes both theoretical guarantees and practical algorithmic design (FDR + PDC), showing an attempt to bridge theory and implementation.
- The idea of addressing spurious correlations or feature shifts in OOD detection could be valuable for improving generalization and reliability.

**Weaknesses:**

- My major concern is that while the paper frames its problem as “Fair OOD Detection” and introduces sensitive attributes to define fairness confusion, the issue it addresses is fundamentally different from conventional fairness concerns (in the fairness literature, the fairness issue typically refers to concerns about minority groups defined by sensitive attributes such as gender, race, or age.). The so-called sensitive attributes, such as image background, are better understood as sources of spurious correlations that affect OOD generalization rather than attributes tied to social fairness (e.g., race or gender) and the proposed method (PACT) primarily mitigate the impact of these spurious correlations on OOD detection, improving model generalization across distribution shifts. Therefore, the work is more accurately interpreted as addressing OOD robustness under spurious correlations rather than true fairness.
- The second point is the experimental setup does not align with established fairness research. The datasets used are not fairness-related, they do not involve socially sensitive or legally protected attributes such as race, gender, or age, which are typically central to fairness evaluation. Instead, the paper treats non-social attributes (e.g., image background) as “sensitive attributes,” which conceptually correspond more to spurious correlations or contextual biases rather than true fairness concerns. While the authors report results on fairness metrics like DP, EO, and EOD, applying these criteria in such settings may be misleading, as it conflates distributional robustness with social fairness.

**Questions:**

- The main loss (Eq. 13) contains four terms. Why doesn’t the paper include any ablation studies to demonstrate the effectiveness of each term? (showing only the effects of tuning the parameters $\alpha$ and $\beta$ is not sufficient.)
- Is fairness the right conceptual lens here, or would this be better framed as robustness to spurious correlations?
- How does “background” qualify as a sensitive attribute under the fairness framework?

---

### Official Review · Reviewer_5acM · 2025-10-25

**Soundness:** 3
**Presentation:** 4
**Contribution:** 2
**Rating:** 4
**Confidence:** 4

**Summary:**

The paper shows how to make OOD detection fair: it adds a new score and a training method so models catch unseen data without treating any group worse.

**Strengths:**

pro:
Good experiment;

End-to-end design as a framework;

Details for replication;

**Weaknesses:**

con:
Fairness problems typically arise from biased outcomes or mislabeling, for example, women may be 50% of applicants but only 10% of hires, not because those cases are unseen. Framed this way, the paper reads more like a class-imbalance/selection-bias treatment than an OOD issue.

The core challenge is that problematic data are heterogeneous—you can’t tell them apart by just using part of the data. As a result, the paper leans on hand-tuned parameters (or hand-picked training sets) to label OOD vs. normal, which undermines the objectivity that fairness demands.

If we already know the sensitive feature, why not just ignore it when we compute OOD? Maybe unawareness fairness, proxy features? I think the paper needs to investigate more on the use case.

**Questions:**

see above

---

### Official Review · Reviewer_RRg2 · 2025-10-30

**Soundness:** 3
**Presentation:** 3
**Contribution:** 3
**Rating:** 4
**Confidence:** 4

**Summary:**

This paper presents the first study on constructing a fairness-aware OOD detection framework, which incorporates the consideration of the fairness metric into OOD detection. To address this challenge, the study introduces a novel metric, Fair-OOD, to identify the Fairness Confusion issue in OOD detection. Furthermore, it proposes a novel algorithm, the Predictive Adaptive Calibration, which is a theoretically guaranteed semi-supervised solution. Extensive experiments on real-world datasets demonstrate that PACT effectively improves OOD detection performance while simultaneously mitigating bias and resolving unfairness.

**Strengths:**

o	The Clear Definition of Fairness Confusion (FC) and the Approach to its Two Contributing Factors. The paper clearly defines Fairness Confusion (FC), which is the bias induced by sensitive attributes and their induced Feature Shifts (FS). The approach is strong as it introduces the challenging problem of Fair OOD Detection by simultaneously considering both the sensitive attributes and the Feature Shifts.
o	The Meaningful proposal of the Fair-OOD Metric. The proposed Fair-OOD metric is significant because it accounts for both sensitive attributes and Feature Shifts, explicitly addressing factors overlooked by conventional fairness metrics. Existing fairness metrics fail to reliably detect unfairness in OOD detection due to their inability to address FC induced by FS. Fair-OOD is proven effective as it can identify FC issues in models that existing fairness metrics fail to detect.
o	The Effective Solution through FDR and PDC terms. The paper proposes the Predictive Adaptive Calibration (PACT) algorithm. It utilizes two key components designed for specific roles:
	Feature Distribution Regularization (FDR): This term constrains the model to extract highly compact feature representations, encouraging it to focus exclusively on class-related features while disregarding sensitive attributes and their induced shifted features, thus demonstrating debiasing capability.
	Predictive Distribution Calibration (PDC): This term mitigates the FC issue by maximizing the divergence (discrepancy) between the prediction distributions of ID and OOD data. Theoretical guarantees confirm that optimizing PDC reduces the impact of FS on prediction distributions, mitigating induced FC.
o	Verification of Superiority through Extensive Experimental Results. Extensive experiments on real-world datasets confirm the method's superiority. The results demonstrate that PACT effectively improves OOD detection performance while simultaneously eliminating both FC and unfairness issues. Furthermore, PACT not only mitigates bias and resolves FC but also achieves substantially closer ACC-U and ACC-B values across different sensitive attributes, indicating robust performance.

**Weaknesses:**

o	It remains uncertain whether the method’s effectiveness will be robustly maintained in scenarios involving multi-categorical sensitive attributes, necessitating additional verification.
o	It can be questioned whether the threshold-based binary classification method used to determine Feature Shift is sufficient for capturing subtle variations in features.
o	The lack of a theoretical justification or empirical evidence for using fixed ID to OOD class split ratios across datasets, such as the 7:3 ratio applied to CIFAR-10-C and ImageNet-100-C, constitutes a weakness.
o	The practice of using the exact same set of classes for ID and OOD partitions across every experimental run seems less convincing.
o	It is less convincing to use a simple binary classification based merely on the presence or absence of corruption, especially since datasets like CIFAR-10-C are composed of diverse noise types (e.g., Gaussian noise, blurs). This simplification, while facilitating formal analysis, potentially masks critical differences in Fairness Confusion (FC) caused by specific corruption categories.
o	The proposed methodology is primarily geared toward fair OOD detection. It is questionable whether its effectiveness can be maintained and verified on existing, generalized OOD benchmarks that lack sensitive attribute definitions.

**Questions:**

o	Why are the sensitive attribute and feature shift binary? In actual fairness research, sensitive attributes often have multiple properties. And also curious why feature shift was determined simply as binary. Is binary classification using a threshold actually more effective at representing feature change?
o	Is there an ablation study that investigates the resulting performance changes when this ID:OOD class ratio is varied?
o	Wouldn't changing the specific classes designated as ID and OOD (e.g., using different class combinations for ID/OOD split) for each experiment lead to a more fair and generalizable result for the same dataset?
o	Given the authors' claim that the framework can be easily generalized to categorical sensitive attributes, and assuming the current binary setup was chosen for 'convenience in formal analysis,' what results would be expected, or obtained, if the experimental setup were complicated to evaluate performance across multi-categorical attributes defined by specific corruption types?
o	Given that the proposed PACT method is a semi-supervised solution, can it still be effectively applied to standard OOD datasets where the OOD samples do not have sensitive attribute labels?
o	Which dataset and setup were used for the ablation study conducted in Section 5.3?

---

### Official Review · Reviewer_o9ip · 2025-11-01

**Soundness:** 2
**Presentation:** 2
**Contribution:** 2
**Rating:** 2
**Confidence:** 5

**Summary:**

This paper introduces Fair OOD Detection in this paper, which simultaneously considers OOD detection and bias induced by Fairness Confusion (FC) caused by sensitive attributes and their induced Feature Shifts (FS). Furthermore, they propose a metric termed
 Fair-OOD to identify FC phenomena in OOD detection, and a theoretically guaranteed semi-supervised solution named Predictive Adaptive Calibration (PACT) to simultaneously enhance OOD detection capability, ensure fairness, and mitigate FC without requiring the label of sensitive attribute for OOD data.

**Strengths:**

- consider fairness and out-of-distribution simultaneously.
- introduce two regularizers -- predictive distribution calibration and predictive distribution calibration.

**Weaknesses:**

- This paper is a combination of fairness and out-of-distribution, which is through two regularizers. The novelty is low.
- The introduction and definitions are 4+ pages. The method about this paper is about 1 page.

**Questions:**

- The FDR term may not be appropriate. When $x^I_j\in D^+$, it is acceptable; however, outside this domain, the term tends to $\infty$. The minimization of the FDR term is for what?
- In FDR term, all $x^I_j$ should compare with some $x^+$, not all?
- In Definition 1, what about $S'_a,S'_{a'}$? Are there 4 possible choices in feature shift $S$?

---

### Meta-Review · Area_Chair_L21j · 2025-12-17

**Summary:**

While the reviewers agree the paper introduces fair OOD detection, there are significant concerns on the novelty, problem setup, technical details, experiments, and presentation. All the reviewer scores are on the negative side, and the authors did not post any rebuttals, which means that they agree with the reviews. This is a clear signal for rejecting the paper this time, and we hope that the comments can help further improve the paper.

**Reviewer Concerns:**

The authors did not respond to the reviewer concerns.

**Reviewer Scores:**

The reviewers would not have changed their scores as there was no discussion.

---

### Decision · Program_Chairs · 2026-01-26

Reject